# Identification of SARS-CoV-2 Main Protease Inhibitors Using Chemical Similarity Analysis Combined with Machine Learning

**DOI:** 10.3390/ph17020240

**Published:** 2024-02-12

**Authors:** Karina Eurídice Juárez-Mercado, Milton Abraham Gómez-Hernández, Juana Salinas-Trujano, Luis Córdova-Bahena, Clara Espitia, Sonia Mayra Pérez-Tapia, José L. Medina-Franco, Marco A. Velasco-Velázquez

**Affiliations:** 1DIFACQUIM Research Group, School of Chemistry, Universidad Nacional Autónoma de México, Mexico City 04510, Mexico; 2School of Medicine, Universidad Nacional Autónoma de México, Mexico City 04510, Mexico; 3Graduate Program in Biomedical Sciences, Universidad Nacional Autónoma de México, Mexico City 04510, Mexico; 4Research and Development in Biotherapeutics Unit (UDIBI), National School of Biological Sciences, Instituto Politécnico Nacional, Mexico City 11350, Mexico; 5National Laboratory for Specialized Services of Investigation, Development and Innovation (I+D+i) for Pharma Chemicals and Biotechnological Products, LANSEIDI-FarBiotech-CONACHyT, Mexico City 11350, Mexico; 6National Council of Humanities, Science and Technology (CONAHCYT), Mexico City 03940, Mexico; 7Immunology Department, Institute for Biomedical Research, Universidad Nacional Autónoma de México, Mexico City 04510, Mexico; 8Immunology Department, National School of Biological Sciences, Instituto Politécnico Nacional, Mexico City 11350, Mexico

**Keywords:** chemoinformatics, COVID-19, SARS-CoV-2, Mpro, 3CLpro, virtual screening, ligand-based drug discovery

## Abstract

SARS-CoV-2 Main Protease (Mpro) is an enzyme that cleaves viral polyproteins translated from the viral genome, which is critical for viral replication. Mpro is a target for anti-SARS-CoV-2 drug development. Herein, we performed a large-scale virtual screening by comparing multiple structural descriptors of reference molecules with reported anti-coronavirus activity against a library with >17 million compounds. Further filtering, performed by applying two machine learning algorithms, identified eighteen computational hits as anti-SARS-CoV-2 compounds with high structural diversity and drug-like properties. The activities of twelve compounds on Mpro’s enzymatic activity were evaluated by fluorescence resonance energy transfer (FRET) assays. Compound **13** (ZINC13878776) significantly inhibited SARS-CoV-2 Mpro activity and was employed as a reference for an experimentally hit expansion. The structural analogues **13a** (ZINC4248385), **13b** (ZNC13523222), and **13c** (ZINC4248365) were tested as Mpro inhibitors, reducing the enzymatic activity of recombinant Mpro with potency as follows: **13c** > **13** > **13b** > **13a**. Then, their anti-SARS-CoV-2 activities were evaluated in plaque reduction assays using Vero CCL81 cells. Subtoxic concentrations of compounds **13a**, **13c**, and **13b** displayed in vitro antiviral activity with IC_50_ in the mid micromolar range. Compounds **13a**–**c** could become lead compounds for the development of new Mpro inhibitors with improved activity against anti-SARS-CoV-2.

## 1. Introduction

SARS-CoV-2 is an enveloped single-stranded positive RNA virus with significant economic and health repercussions [1,2,3]. This virus, identified at the end of 2019, has a genetic similarity of 79–96.3% with other coronaviruses [1,4]. SARS-CoV-2 infection is mainly mediated by spike (S) protein binding to angiotensin-converting enzyme 2 (ACE2) on the surface of host cells and the subsequent S protein cleavage that activates membrane fusion [5]. The initial 20 kb downstream the 5′ end of the viral genome contains the open reading frames (ORFs) 1a and 1b, which encode the polyproteins 1a and 1ab. Polyprotein 1a (pp1a) comprises nonstructural protein (NSP) 1 to NSP11, whereas polyprotein 1ab (pp1ab) includes NSP1 to NSP16 [1]. The generation of NSPs from pp1a and pp1ab requires the release by the papain-like protease (PLPro) and the Mpro (also known as 3CLPro) by self-cleavage [1,6,7]. NSPs are vital to viral replication [8,9]. For example, RNA-dependent RNA polymerase (RdRp/NSP12) is a crucial component of the genome replication/transcription complex; helicase (NSP13) unravels double-stranded (ds) DNA and RNA along the 5′–3′ direction; and Mpro and PLPro participate in the excision of the pps [10].

Mpro is a cysteine protease with hydrolase activity that cleaves the sequence Leu-Gln↓Ser-Ala-Gly, which is unusual in human enzyme substrates [11]. The bioactive form of Mpro is a homodimer, where the rearrangement of the catalytic pocket in the monomers induced by dimerization promotes substrate recognition [11,12]. Each monomer has three domains. Domains I and III mediate dimerization through a salt bridge between Glu209 and Arg4 of opposite monomers [11,13]. The catalytic pocket and the catalytic residues His41 and Cys145 are in domains I and II [10,14]. The substrate-binding pocket of Mpro has been divided into five sub-pockets—S1, S2, S3, S4, and S1′—which accommodate different residues of the substrate [10,15]. S2 and S4 are hydrophobic cavities, while S1 mediates the recognition of molecules similar to glutamine, and the small S1′ cavity allows the thiol of Cys145 to come into contact with the substrate [13,15].

Mpro has multiple characteristics that make it a prominent therapeutic target for drug development: (i) it is crucial for the viral replicative cycle; (ii) it has a high (96%) nucleotide similarity between coronaviruses; (iii) it has a low mutation rate; and (iv) it has no homology to human proteins [11,16]. Accordingly, multiple Mpro inhibitors have been developed to date. For example, boceprevir, SY110, and GC376 have been effective in preclinical studies [17,18,19]; simnotrelvir, ensitrelvir, ebselen, and masitinib have reached clinical studies [20,21,22,23]; and nirmatrelvir has been approved by regulatory agencies for the clinical treatment of non-severe COVID-19 [11,24]. Nirmatrelvir decreases the hospitalization and death risks by 90%, demonstrating that Mpro inhibition is clinically relevant. The two non-covalent inhibitors, ebselen and masitinib, form π-π interactions with His 41. On the other hand, simnotrelvir and nirmatrelvir form a covalent bond with Cys146 and hydrogen bonds with Glu166. All four inhibitors form hydrogen bonds with His163 [25,26,27,28].

The medicinally relevant chemical space is vast, enabling the identification of candidate compounds for therapeutic development [29,30]. Indeed, the chemical and biological space continues expanding, and it is still challenging to develop and implement efficient methodologies to explore such large and evolving spaces for drug discovery [31]. Virtual screening (VS) or computational filtering of compound collections is a powerful tool for systematically exploring the chemical space that allows the evaluation of some compounds to be prioritized over others in subsequent drug development phases [32]. VS has been used to identify candidate compounds for SARS-CoV-2 molecular targets [33].

VS allows for the analysis of up to billions of small molecules [34,35] since it can use simplified representations of chemical entities, overcoming the main limitations of hit identification based only on experimental approaches, such as high-throughput screening [36,37]. VS based on the ligand uses knowledge about the intrinsic characteristics of molecules with defined activity to find new potential active compounds. Those approaches are based on the similarity principle, which states that “a compound structurally similar to active compounds will probably also be active”, with the relevant exceptions of activity and property cliffs (e.g., chemical compounds with very similar chemical structures but very different activity profiles) [37,38]. In particular, ligand-based VS relying on the structural similarity principles typically uses molecular fingerprints generated from two-dimensional representations to describe molecular structures and compares the structural similarity between molecules within a chemical database (e.g., screening collection) and compounds with the desired activity (e.g., reference or queries) through a similarity metric [39,40,41]. The prediction of pharmacokinetic properties at this stage allows for the selection of molecules with a better probability of success and susceptibility to optimization [42,43]. However, identifying new bioactive compounds with the aid of VS still requires the experimental testing of the computationally filtered compounds through in vitro and in vivo assays to identify compounds with value in drug development [33].

Herein, we aimed to identify SARS-CoV-2 Mpro inhibitors using ligand-based VS based on similarity searching. We virtually screened a chemical library with nearly 17 million molecules using compounds with demonstrated activity against coronavirus as references or queries. For the refinement of the computational hits, we used the different machine learning (ML) algorithms available in the Assay Central software, which were previously employed to search for molecules against SARS-CoV-2 [44,45,46], and the algorithm developed by Alves et al. [47] for identifying Mpro inhibitors. One commercially available computational hit (compound **13**) effectively inhibited Mpro in FRET assays. In a follow-up analysis, compound **13** was used to explore its local SAR through the selection and experimental testing of three new analogs that are commercially available (**13a**–**c**). Those four compounds inhibited Mpro enzymatic activity with different potencies, and three of them reduced the cytopathic effect elicited by SARS-CoV-2 infection in Vero CCL81 cell cultures. Our study provides new candidates for developing Mpro inhibitors and anti-SARS-CoV-2 drugs.

## 2. Results and Discussion

### 2.1. In Silico Identification of Potential Mpro Inhibitors

Given the need to find new inhibitors for SARS-CoV-2 Mpro [48,49], we conducted a VS using the strategy summarized in Figure 1. First, from the literature, we identified anti-coronavirus compounds with diverse mechanisms of action (Appendix A) to be used as references for similarity searching from a compound screening library with ~17 million molecules of bioactive compounds, natural products, and peptides [50,51,52,53,54,55,56,57,58]. Of the 62 reference molecules, thirteen (Appendix A) have been shown to inhibit at least one of two SARS-CoV-2 proteases [59,60,61,62,63,64,65,66,67,68,69,70]. Using three different molecular fingerprints, we identified 632,149 unique molecules that had similarity values greater than the mean plus three standard deviations (SDs) for all three of the molecular fingerprints. We verified the availability of physical samples of these molecules in the ZINC^15^ database, finding 4423 molecules listed as commercially available. This group of compounds includes candidates that may have anti-coronavirus activity independently of its mechanism of action. Our primary selection, using fingerprint-based similarity searching, has been successfully employed in VS campaigns to identify inhibitors of the protein kinase AKT-2 [71] and, more recently, a novel inhibitor of the epigenetic target DNA methyltransferase 1 [72].

To prioritize the evaluation of a smaller number of compounds, we decided to focus on the identification of potential Mpro inhibitors by implementing the seven ML algorithms available in the Assay Central Software [44] and the ML-QSAR of Alves et al. [47], which have successfully identified SARS-CoV-2 Mpro inhibitors [44,45,46,47]. This further refinement directed us to select potential Mpro inhibitors, defining the subsequent experiments. With the overall strategy, we identified eighteen compounds as computational hits (Appendix A).

### 2.2. Property and Diversity Profiling of Computational Hits and Experimental Validation

The computational hits chosen by our protocol (Figure 2a and Appendix A) had an overall high structural diversity with an average Tanimoto coefficient < 0.2 computed with the ECFP6 fingerprint (Figure 2b). However, we identified two groups of molecules that were structurally related: compounds **5** and **6**, and compounds **14**–**18**. The predicted physicochemical, pharmacokinetic, and toxicological properties for all computational hits were in the range of typical drug-like molecules, which was expected, given the composition of the reference set. Most of the computational hits complied with seven desired values for properties associated with favored oral bioavailability (water solubility, topological polar surface area (TPSA), molecular weight, rotatable bonds, lipophilicity, and H-bond donors and acceptors) (Figure 2c). However, compounds **5**, **14**, **15**, and **16** had a TPSA < 20 Å^2^, and compound **8** had reduced lipophilicity.

Twelve out of eighteen computational hits were purchased, and their activities on Mpro activity were assessed by FRET analysis at the single concentration of 100 µM. We found that two compounds (**3** and **13**) significantly inhibited Mpro compared to their vehicle control (Figure 2d). Compound **13** was the most active, inhibiting >90% Mpro activity, demonstrating a successful large-scale VS.

### 2.3. Identification of New Mpro Inhibitors by Searching Analogs of Compound ***13***

We selected and experimentally tested analogs of compound **13**. We identified three compounds (Figure 3a) that were commercially available and had high similarity to compound **13**: **13a** (ZINC4248385), **13b** (ZINC13523222), and **13c** (ZINC4248365) (Figure 3b and Appendix A). The newly identified analogs had desired physicochemical properties associated with favored oral bioavailability (Figure 3c), complying with Lipinski’s and Verber’s rules, and showing high (>86%) predicted absorption after oral administration (Table 1). None of the compounds were predicted to cross the blood–brain barrier (BBB). Compounds **13** and **13a**, but not **13b** or **13c**, may inhibit multiple CYP isoforms and cause hepatotoxicity. Our analysis indicates that compound **13c** may have an extended half-life in humans (Table 1).

The dose–response curves with recombinant SARS-CoV-2 Mpro showed that the positive control, GC376, inhibited Mpro with a high potency (Appendix A). Compounds **13**, **13b**, and **13c** inhibited the enzymatic activity of Mpro with IC_50_ values of 3.5, 29.1, and 1.8 µM, respectively (Figure 3d). The inhibitory potency of compounds **13**, **13b**, and **13c** is similar to other compounds identified using other VS strategies. For example, docking and subsequent molecular dynamics found Mpro inhibitors with IC_50_ values ranging from 6.74 to 1370 μM [26].

SAR expansion has been employed for the optimization of protease inhibitors with antiviral activity. For example, benserazide analogs identified by SAR expansion display increased affinity for the 3C protease from human coxsackievirus B3, and improved antiviral activity [73]. Herein, the SAR expansion of compound **13** allowed for the identification of two new Mpro inhibitors, and one of them (**13c**) had a 1.9-fold increase in potency.

### 2.4. In Silico Characterization of Compound Binding to SARS-CoV-2 Mpro

Compounds **13**–**13c** were docked into the catalytic cavity of SARS-CoV-2 Mpro. Analysis of the best-ranked poses of compounds **13**–**13c** into SARS-CoV-2 Mpro showed that, as expected given the high structural similarity of the compounds, they have similar binding modes, all occupying sub-pockets S1 and S2 of the active site (Figure 4a). The docking poses suggested π-π interactions between the pyridone motif of compounds **13a**–**13c** and the side chain of His41 (Figure 4b). Residues Cys145 and His41 are essential for the proteolytic activity of SARS-CoV-2 Mpro since they constitute its catalytic dyad [13,15]. Thus, ligands interacting with Cys145 and/or His41 reduce SARS-CoV-2 Mpro activity. For example, Mpro covalent inhibitors target Cys145 [11,17,18].

According to the predicted docked poses, other residues potentially mediating the binding of compounds **13**–**13c** to Mpro included the polar residues Gly143 and His163, the hydrophobic residue Leu141, and the negatively charged residue Glu166 (Appendix A). The interactions with His163 and Glu166 may participate in the enzymatic inhibition reported here since they would affect substrate binding. During catalysis, Mpro His163 forms a hydrogen bond with the conserved Gln adjacent to the cleavage site, and Glu166 forms extensive van der Waals interactions and hydrogen bonds with multiple atoms from the substrate [74]. Importantly, Cys145, His163, and Glu166 are conserved among coronavirus Mpro [75], and their frequency of mutation in circulating SARS-CoV-2 variants has been reported to be <1 per million of sequences by two different studies [75,76]. Altogether, these data suggest that compounds **13**–**13c** could display broad-spectrum activity against coronavirus, a desirable characteristic for pandemic preparedness [77].

We did not find a correlation between the docking scores for compounds **13**–**13c** (Appendix A) and their inhibitory activities. In the molecular dynamics simulation of the **13c**-SARS-CoV-2 Mpro complex, the ligand rapidly abandoned the docked conformation and explored multiple conformations inside of the catalytic cavity. Eventually, ligand **13c** moved outside of the cavity because of the relocation of a highly flexible loop comprising Mpro residues 43 to 54 (Appendix A). Together, these results suggest that our docking results may not accurately capture the molecular interactions that influence the observed inhibitory activity. This can be caused by the inherent limitations of docking methods, including approximations in representing isolated systems and the omission of factors like solvation [75,76], combined with the complexity of the biological target. In agreement, multiple docking protocols for Mpro have shown unsatisfactory performance [39,78]. Thus, clarification of the binding modes of compounds **13**–**13c** will require the generation of experimental structures in future work.

### 2.5. Evaluation of Antiviral Activity in Cell Culture

First, we determined the effect of compounds **13–13c** on the viability of Vero CCL81 cells (Appendix A). The exposure of cell cultures to the reported Mpro inhibitor GC376 for 72 h reduced cell viability with respect to the vehicle, but the reduction was greater than 20% only at the highest concentration tested (100 µM). Compounds **13**, **13a**, **13b** (0.78–100 µM), and **13c** (0.39–50 µM) had null cytotoxicity compared to their corresponding vehicles (DMSO 0.2% for **13**–**13b** and DMSO 0.5% for **13c**). Determining the cytotoxic effect of the test compounds is essential to ensure a healthy cell layer during antiviral evaluation, reducing false positive results, and allowing for speculation that the antiviral activity could be achieved in vivo without toxic effects. Thus, the FDA guidance documents for the development of antiviral products define cytotoxicity evaluation as crucial [79].

Using subtoxic concentrations of the Mpro inhibitors **13**–**13c**, we performed plaque reduction assays in cell cultures infected with SARS-CoV-2 (see Appendix A for representative images). The positive control, GC376, inhibited lytic plaque formation (Figure 5a) with an EC_50_ value that was consistent with that of previous reports [18,80]. Surprisingly, compound **13** lacked antiviral activity in vitro (Figure 5b) despite its efficient inhibition of Mpro in FRET assays. The discrepancy between enzymatic inhibition and antiviral efficacy could be caused by a reduced stability of the compound in cell culture, permeability issues, increased efflux transport in Vero cells, or buffering by host or viral off-target proteins. Compound **13a** showed the best antiviral effect among the candidates with an EC_50_ value of 35.3 µM (Figure 5c). Compounds **13b** and **13c** displayed EC_50_ values of 59.9 and 57.0 µM, respectively (Figure 5d,e). The anti-SARS-CoV-2 activity of compounds **13a**–**c** confirms that they could be new lead compounds for the development of Mpro inhibitors.

## 3. Materials and Methods

### 3.1. Preparation of the Screening Chemical Library and the Reference Set of Active Compounds

We generated a large chemical library with 17,802,481 compounds from eight chemical libraries (dataset comprising CAS COVID-19 candidate compounds: 44,787 compounds; ChEMBL version 25: 1,667,509 compounds; COlleCtion of Open NatUral producTs (COCONUT): 379,309 compounds; the Food Database (FooDB): 23,883 compounds; the Colombian database of Chemistry of Bioactive Plant Natural Products (CCBPNP in-house database): 535 compounds; REadily AccessibLe database (REAL) by Enamine: 15,547,017 compounds; Dark Chemical Matter (DCM) database: 139,352 compounds; and the peptide FDA-approved drug (PEP FDA): 92 compounds. The large chemical library was curated using an in-house protocol described in detail in the literature [81]. Briefly, from the Simplified Molecular Input Line Entry Specification (SMILES) notation of each molecule [82], their largest fragment was kept. Duplicate molecules and those that contained atoms other than H, B, C, N, O, F, Si, P, S, Cl, Se, Br, and I were dismissed. Then, the corresponding tautomers were generated through the neutralization and reionization, implementing public tools and an application programming interface (API) in Python based on the RDKit toolkit for cheminformatics. The API uses the “Standardizer”, “LargestFragmentChoser”, “Uncharger”, “Reionizer”, and “TautomerCanonicalizer” functions from the MolVS standardize and validation tools section. In parallel, we built a reference list of sixty-two molecules with anti-coronavirus activity reported until 2021 [50,51,52,53,54,55,56,57,58] (Appendix A), which was prepared in the same way as the large chemical library and used in the first phase of the VS.

### 3.2. VS

We developed an API in Python using the “Fingerprinting and Molecular Similarity”, “Topological Fingerprints”, “MACCS Keys”, and “Morgan Fingerprints (Circular Fingerprints)” modules from the RDKit toolkit for cheminformatics. The molecules in the large chemical library and the reference list were represented using three 2D molecular fingerprints of different designs: Extended Connectivity Fingerprints of radius two and three (ECFP4 and ECFP6)—2048 bits [83]—and Molecular Access System Keys (MACCS Keys)—166 bits [84]. The structural similarity between the molecules was determined using the Tanimoto coefficient [84,85]. We identified molecules with similarity values greater than the average similarity plus three SDs for each reference molecule. Those unique molecules above the cutoff limit from all three molecular fingerprints were considered consensus hits, and their commercial availability was determined using the ZINC^15^ database using the following keywords: “For sale”, “In Stock” and “Now”. Refinement of the computational hits included using the different ML algorithms available in the Assay Central Software [44] (available at https://www.collaborationspharma.com/assay-central (accessed on 20 December 2023)) as well as the quantitative structure–activity relationship based on machine learning (ML-QSAR) using a random forest algorithm developed by Alves et al. [47] to search for Mpro inhibitors. The lists of reference molecules employed in those algorithms have been previously presented [44,45,46,47].

The ML algorithms of Assay Central software employs Bayesian ML models, including but not limited to Bernoulli Naive Bayes classifiers, Adaptive Boosting, and Random Forest, all of which have been previously used to search for molecules against SARS-CoV-2 [44,45,46]. These algorithms operate based on the result obtained with the molecular descriptor ECFP6, providing a comprehensive assessment of predictability and applicability. Molecules with scores greater than 0.5 in these predictions are identified as active.

The ML-QSAR algorithm integrates Random Forest classifiers and three distinctive descriptors: Morgan fingerprints, SiRMS, and Dragon version 7.0. This comprehensive model evaluates various features of compounds, including lipophilicity, partial charges, refractive index, and the potential to form hydrogen bonds (as acceptors or donors). The results of the ML-QSAR model undergo scrutiny through an average consensus analysis of the three descriptors, with the activity of each prediction being scaled from zero to one. A consensus result exceeding 0.7 (70%) designated compounds as positive hits, and results were categorized as either 0 (negative) or 1 (positive).

Compounds with favorable predictions in both the ML algorithms and ML-QSAR predictions were regarded as computational hits, contributing to a refined selection of molecules with promising activity against SARS-CoV-2 Mpro.

### 3.3. Analysis of Structural Diversity and Physicochemical Properties of Computational Hits

The Tanimoto coefficient for ECFP6 fingerprints was used to measure the similarity between the computational hits. Clustering maps were generated using the DisplayR software version 1.2.37478 (Displayr, Pyrmont, Australia). Physicochemical properties for the final computational hits were calculated using the SwissADME [86] and pkCSM-pharmacokinetics web servers [87].

### 3.4. Compounds

Compounds were purchased from MolPort (Molport, SIA, Riga, Latvia), Vitas M Chemical (Hong Kong, China), Sigma-Aldrich (St. Louis, MO, USA), Enamine (Kyiv, Ukraine), TargetMol (Wellesley Hills, MA, USA), MedChemExpress (Monmouth Junction, NJ, USA), or Life Chemicals (Niagara-on-the-Lake, ON, Canada) for evaluation of their activity on Mpro enzymatic activity, cell viability, and SARS-CoV-2 replication in cell culture. Details of vendors and catalog numbers can be found in Online Resource 2. The reported Mpro inhibitor GC376 [18] was obtained from Cayman Chemical (Cat. 31469). All compounds were dissolved in dimethyl sulfoxide (DMSO).

### 3.5. Mpro Enzymatic Activity Assays

Mpro activity was evaluated using a continuous kinetic FRET assay performed according to the protocol and specifications of the supplier Reaction Biology Corp. (Malvern, PA, USA) or Charles River Laboratories (Saffron Walden, Essex, UK). Briefly, the compounds were mixed with recombinant Mpro in the reaction buffer (Tris 25 mM pH 7.3, EDTA 1 mM, Triton X-100 0.005%). Protease activity was monitored as a time-course measurement of the increase in fluorescence signal (excitation 340 nm and emission 492 nm) after the addition of the fluorogenic substrate [NH2-C(EDANS)VNSTQSGLRK(DABCYL)M-COOH]. Initial enzymatic rates were calculated via linear regression using data from the initial proportion of the kinetic curve. Data were normalized using enzymatic rates from vehicle control, and control without enzyme as maximal and minimal responses, respectively. GC376 was employed as a positive control. Dose–response curves were performed for a selected subset of compounds with nine concentrations (3-fold serial dilution from 100 μM). Non-linear regression and IC_50_ values were obtained using the Prism 10 v 10.1.1 (GraphPad software, Boston, MA, USA). Experiments were repeated two times independently.

### 3.6. Molecular Docking

Docking simulations for selected compounds (**13**–**13c**) were performed with the Molecular Operating Environment (MOE) software, version 2018 (Chemical Computing Group ULC, Montreal, Canada), using the crystal structure of SARS-CoV-2 Mpro retrieved from Protein Data Bank (PDB ID: 7JPZ) [88] and the induced fit protocol. Before docking simulations, the co-crystallized inhibitor structure was excised from the catalytic site. Protein preparation within the MOE framework involved using the “Quick prepare” module, employing default parameters and the AMBER10:EHT force field. The chemical structures of compounds **13**–**13c** were constructed and prepared in MOE starting from their canonical SMILES. The preparation process included the application of the “wash”, “partial charges” calculations, and “energy minimize” functions to optimize ligand conformations. The molecular docking simulations were executed utilizing the “Triangle Matcher” method, complemented by the “London dG” function, with a maximum of 1500 iterations and an initial population of 100 poses.

### 3.7. Molecular Dynamics Simulations

Molecular dynamics (MD) simulation was conducted using GROMACS 2021.6 with the CHARMM36m force field [89]. The complex of **13c** with SARS-CoV-2 Mpro was obtained with the docking protocol described above. Ligand parameters were generated through the CHARMM graphical user interface [90] module within the force field framework. The complex was placed in a periodic cubic simulation box measuring 99 Å on each side, followed by solvation using the three-point (TIP3P) model to incorporate water molecules. All histidines were protonated as neutral HSD states. Sodium (Na+) and chloride (Cl^−^) ions were added to neutralize the system charge and reach an ionic concentration of 0.15 M. Energy minimization utilized the steepest descent algorithm, followed by equilibration in an NVT ensemble employing a modified Berendsen thermostat. The LINCS algorithm was applied to constrain hydrogen bonds. Simulations were run at 1 bar and 310.15 K for 250 ns with a 2 fs integration time frame, saving trajectories every 10 ps.

### 3.8. Cytotoxicity Assessment

Vero CCL81 cells were seeded in 96-well plates at 12,000 cells/cm^2^ in Eagles’s Minimum Essential Medium (EMEM) supplemented with 10% fetal bovine serum (FBS). Compounds (2-fold serial dilutions from 100 µM) were added to the wells and incubated for 72 h at 37 °C with 5.0% CO_2_. Vehicle control was DMSO 0.2%, except for **13c**, which was compared to DMSO 0.5%. The cytotoxic drug doxorubicin (5 µM) was used as a positive control. Cell viability quantification was performed using 3-(4,5-Dimethylthiazolyl)-2,5-diphenyltetrazolium bromide (MTT) as previously reported [91]. Data were normalized with respect to the corresponding vehicle control.

### 3.9. Evaluation of Antiviral Activity by Plaque Reduction Assay

In vitro viral neutralization assays were performed as previously described [92] using a clinical isolate of SARS-CoV-2 (GenBank: OL790194). Briefly, cultures of Vero CCL81 cells (50,000 cells/cm^2^) were seeded into 24-well plates, and the next day, the cells were exposed for 1 h to subtoxic concentrations of the Mpro inhibitors prepared in serum-free EMEM. Viral infection was performed by adding 100 SARS-CoV-2 plaque-forming units for 1 h. Then, the cell culture medium was replaced by an overlay media (carboxymethyl cellulose 1% *w*/*v*, FBS 1% *v*/*v*, in EMEM) containing the same concentration of the test compounds. GC376 was used as the positive control, and uninfected cultures were used as negative controls. At the end of the incubation period, the cell cultures were fixed with formaldehyde 37% and incubated overnight at 4 °C. Then, the medium was removed, and the cell layer was stained with crystal violet. Finally, the lytic plaques were counted, and data were normalized against vehicle controls. The effective concentration-fifty was calculated by non-linear regression with Prism 10 v 10.1.1 (GraphPad software, Boston, MA, USA).

## 4. Conclusions

In this study, we identified three small molecules (**13a**, **13c**, and **13b**) with in vitro antiviral activity with IC_50_ values in the mid micromolar range. The three compounds could become lead compounds for the development of Mpro inhibitors with improved activity against anti-SARS-CoV-2. A ligand-based VS of a large screening library led to the rapid identification of compound **13** with significantly inhibitory activity against SARS-CoV-2 Mpro. A successful follow-up exploration of analogs of the confirmed computational hit led to the finding of two additional active compounds. This study represents a further example of a successful large-scale VS followed by an experimental validation to identify active compounds for a therapeutic relevant molecular target.

## Figures and Tables

**Figure 1 pharmaceuticals-17-00240-f001:**
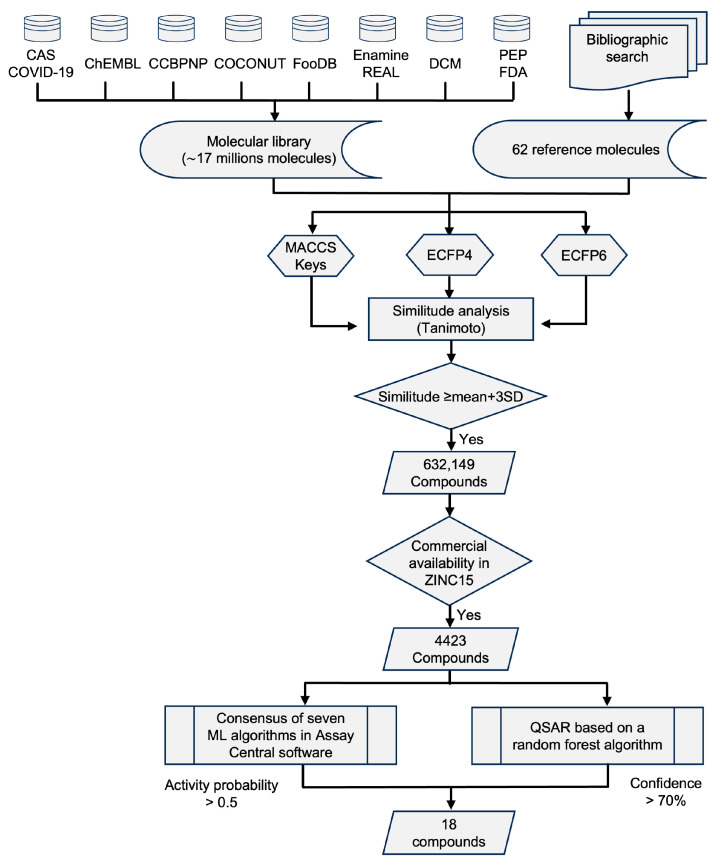
Virtual screening protocol. Identification of potential Mpro inhibitors from a large chemical library combining multiple databases. Similarity analysis was performed with the Tanimoto coefficient and three structural fingerprints (indicated in the figure): MACCS Keys, ECFP4, and ECFP6. ML: machine learning; QSAR: quantitative structure–activity relationship.

**Figure 2 pharmaceuticals-17-00240-f002:**
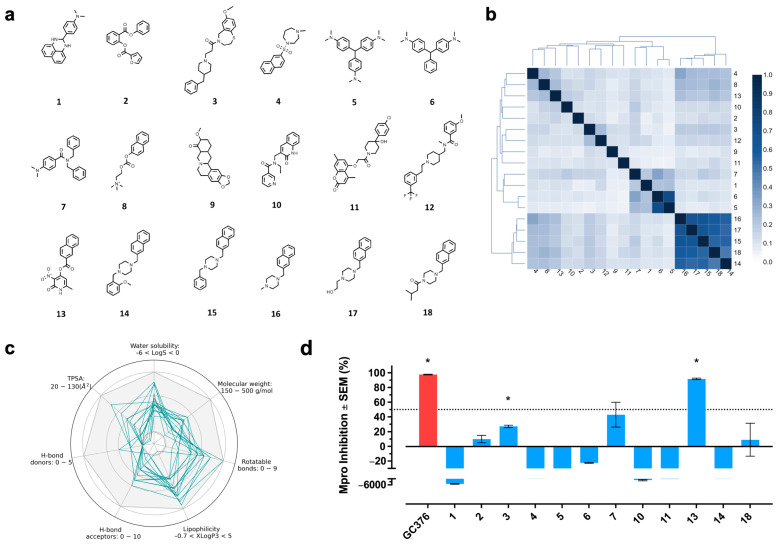
Structure and physicochemical properties of the computational hits. (**a**) Chemical structure of compounds **1**–**18**, identified by our protocol as potential SARS-CoV-2 Mpro inhibitors. (**b**) Similarity matrix of the computational hits computed with the Tanimoto coefficient and the ECFP6 fingerprint. (**c**) Physicochemical properties profiles of the computational hits. The gray area represents the range of values for each property where oral bioavailability is favored. (**d**) Mpro activity in the presence of 100 µM of the selected computational hits (blue bars) or the positive control GC376 (1 µM; red bar) [18]. The signal was normalized and statistically compared against the corresponding vehicle (Student’s *t* test; * < 0.05 for inhibitory compounds). Bars show the average ± standard error of the mean (SEM) from two independent experiments.

**Figure 3 pharmaceuticals-17-00240-f003:**
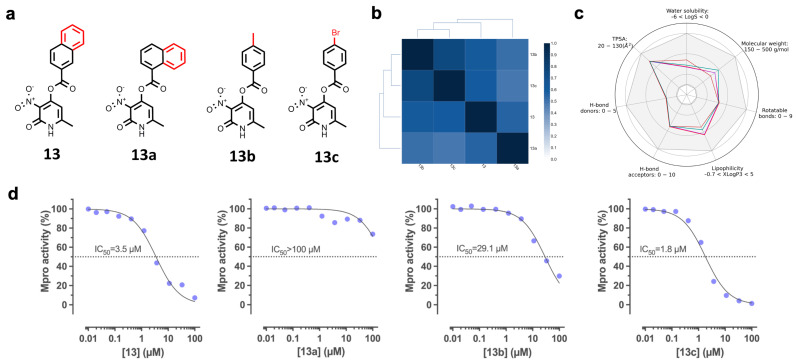
Inhibitory activities of compounds **13**–**13c** on Mpro activity. (**a**) Chemical structure of compound **13** and its three commercially available structural analogs. The maximum common substructure is shown in black. (**b**) Similarity matrix of compounds **13**–**13c** calculated with the Tanimoto coefficient and the ECFP6 fingerprint. (**c**) Physicochemical properties of the compounds (**13**: red; **13a**: purple; **13b**: brown; **13c** green). The gray area represents the range of values where oral bioavailability is favored. TPSA: topological polar surface area. (**d**) Representative concentration–response curves showing the inhibitory activities of compounds **13**–**13c** on Mpro activity (**d**). Two independent assays were performed for each compound.

**Figure 4 pharmaceuticals-17-00240-f004:**
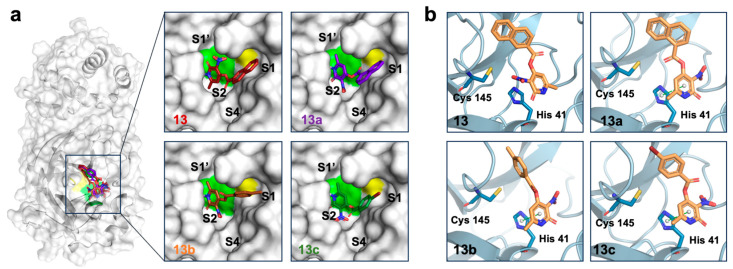
Binding models of Mpro inhibitors generated by molecular docking employing Molecular Operating Environment software. (**a**) Surface representation of monomeric Mpro with compounds **13**–**13c** docked into the catalytic site. The insets show each compound bound the sub-pockets S1 and S2. The catalytic residues His41 and Cys145 are shown in green and yellow, respectively. (**b**) Ribbon representation of the Mpro sub-pocket S2 showing the predicted π-π interactions between the pyridone of compounds **13a**–**13c** and His 41.

**Figure 5 pharmaceuticals-17-00240-f005:**
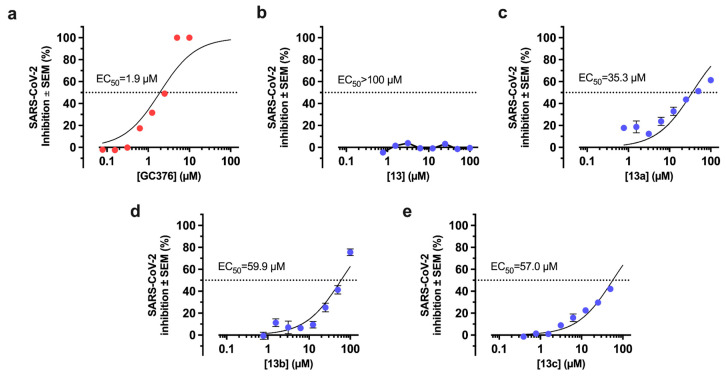
Anti-SARS-CoV-2 effect of Mpro inhibitors. Quantification of the cytopathic effect elicited by SARS-CoV-2 in Vero CCL81 after 72 h in the presence of (**a**) the positive control, GC376; (**b**) compound **13**; or (**c**–**e**) analogs **13a**–**c**. The maximal concentration tested for each compound was determined by evaluating its cytotoxicity in uninfected cells. Graphs show the mean percent reduction in the cytopathic effect from three technical replicates ± SEM. A representative curve is shown from two independent experiments performed.

**Table 1 pharmaceuticals-17-00240-t001:** Pharmacokinetic and toxicology properties of compounds **13**–**13c** predicted using SwissADME and pkCSM-pharmacokinetics.

Molecular Descriptor	Compound 13	Compound 13a	Compound 13b	Compound 13c
Intestinal absorption (%)	89.21	93.79	86.80	87.54
Lipinski violations	None	None	None	None
Verber violations	None	None	None	None
BBB permeability	No	No	No	No
P-glycoprotein substrate	No	Yes	Yes	No
P-glycoprotein inhibitor	Yes	No	No	No
CYP1A2 inhibitor	Yes	Yes	No	No
CYP2C19 inhibitor	Yes	Yes	No	Yes
CYP2C9 inhibitor	Yes	Yes	No	No
CYP2D6 inhibitor	No	No	No	No
CYP3A4 inhibitor	No	No	No	No
Total clearance (log mL/min/kg)	0.75	0.75	0.79	-0.16
AMES toxicity	Yes	Yes	Yes	Yes
Hepatotoxicity	Yes	Yes	No	No

BBB, blood–brain barrier; CYP, cytochrome P450.

## Data Availability

The data presented in this study are available within the article or Appendix A.

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
