# Peer review of "Identification of SARS-CoV-2 Main Protease Inhibitors Using Chemical Similarity Analysis Combined with Machine Learning"

_pharmaceuticals, 2024, doi:10.3390/ph17020240_

Round 1
Reviewer 1 Report
Comments and Suggestions for Authors
The manuscript describes the role of SARS-CoV-2 Main Protease (Mpro) as an enzyme that cleaves viral polyproteins, which is crucial for viral replication. The study conducted a large-scale virtual screening of over 17 million compounds using multiple structural descriptors and machine learning algorithms. Eighteen compounds with diverse structures and drug-like properties were identified as potential anti-SARS-CoV-2 compounds. Twelve of these compounds were evaluated for their inhibitory effect on Mpro enzymatic activity, and compound 13 (ZINC13878776) showed significant inhibition. Structural analogues 13a (ZINC4248385), 13b (ZNC13523222), and 13c (ZINC4248365) were also tested and showed varying levels of potency as Mpro inhibitors. These compounds were further evaluated for their anti-SARS-CoV-2 activity in vitro and demonstrated antiviral activity at subtoxic concentrations. Although the paper is nicely organized and well written, there was still an issue to be improved.
1. Under normal circumstances, virtual screening of drugs typically requires the use of around three protein structures, from which compounds with high simultaneous scores are selected. This approach helps to avoid significant errors caused by variations in protein structures. However, this study only utilized one protein structure, which is clearly inaccurate.
2. In this study, candidate compounds can be evaluated for ADME/T (absorption, distribution, metabolism, excretion, and toxicity) parameters using online databases to determine if they meet certain criteria for traditional Chinese medicine.
3. Molecular docking is a vital tool for exploring the interactions between an inhibitor molecule and the protein target. In addition, MD simulations allow the protein-ligand complex to be fully relaxed in the solvent environment (i.e., taking into account the protein flexibility, which cannot be fulfilled by the molecular docking process), thereby generating more reliable binding properties. Docking and MD simulations are well-established and common methods to computationally probe the structure and dynamics of biological macromolecules in the CADD studies. The author should use the MD method to explore the dynamic binding process of drugs and targets.
Author Response
We thank the Reviewer for her/his comments and suggestions. Please find below the detailed response to the observations made.
1. Under normal circumstances, virtual screening of drugs typically requires the use of around three protein structures, from which compounds with high simultaneous scores are selected. This approach helps to avoid significant errors caused by variations in protein structures. However, this study only utilized one protein structure, which is clearly inaccurate.
For our structure-based similarity analysis, we employed three different fingerprints and selected compounds with high simultaneous similarity scores. This primary strategy reduces the probability of selecting false-positive hits by fingerprint bias. This information has been highlighted in Figure 1 legend and lines 156-158.
On the other hand, for docking experiments we employed only one structure of SARS-CoV-2 Mpro (PDB ID: 7JPZ) since comparison of the crystal structures of Mpro from different SARS-CoV-2 strains lacks of significant differences (10.1007/s11224-022-02089-6). Thus, for this protein, it is redundant to employ multiple structures. No changes were made to the manuscript.
2. In this study, candidate compounds can be evaluated for ADME/T (absorption, distribution, metabolism, excretion, and toxicity) parameters using online databases to determine if they meet certain criteria for traditional Chinese medicine.
The in silico evaluation of ADMET of compounds 13-13c is presented in Table 1. Our candidates were not obtained from traditional Chinese medicine and therefore, we do not expect to meet related criteria. No changes were made to the manuscript.
3. Molecular docking is a vital tool for exploring the interactions between an inhibitor molecule and the protein target. In addition, MD simulations allow the protein-ligand complex to be fully relaxed in the solvent environment (i.e., taking into account the protein flexibility, which cannot be fulfilled by the molecular docking process), thereby generating more reliable binding properties. Docking and MD simulations are well-established and common methods to computationally probe the structure and dynamics of biological macromolecules in the CADD studies. The author should use the MD method to explore the dynamic binding process of drugs and targets.
We have performed MD simulation with compound 13c. The corresponding methods are described in section 3.7 and the results are presented in lines 255-261 and figure S3.
Reviewer 2 Report
Comments and Suggestions for Authors
General comment: In this work, the authors conducted a large-scale virtual screening of analogs of compounds with proven anti-coronavirus activity on Mpro. Subsequently, the activity of 12 computational hits was tested, and finally, preliminary Structure-Activity Relationship (SAR) studies were conducted on the most potent compound, compound 13.
Comment 1: Actually, several virtual screening studies on Mpro have already been published, and in these studies, nanomolar-level non-covalent inhibitors have been identified (J. Am. Chem. Soc. 2022, 144, 2905−2920; J. Med. Chem. 2022, 65, 6499−6512; ACS Med. Chem. Lett. 2023, 14, 10, 1434–1440). I am not sure about the significance of the current authors searching for analogs of inhibitors for other coronaviruses for virtual screening. Moreover, there has been no in-depth SAR studies conducted on the identified hits to obtain nanomolar inhibitors, and there are no experimental structures to validate the docking poses.
Comment 2: Could the authors summarize the reported antiviral activity of reference compounds with the IC50/EC50 values?
Comment 3: Is there any correlation between the docking scores and the predicted antiviral activity? In this manuscript, I did not find information regarding the docking scores.
Comment 4: Regarding the docking poses of compounds 13-13c, in Figure 4, it is evident that in all compounds, the poses show pyridone within the S2 pocket forming π-π interactions with His41. However, in Figure S2, this seems to be inconsistent, especially in Figure S2(A). Could the authors please provide an explanation for this discrepancy?
Comments on the Quality of English Language
The manuscript might benefit from improvements in terms of English style and clarity.
Author Response
We thank the Reviewer for her/his comments and suggestions. Please find below the detailed response to the observations made.
Comment 1: Actually, several virtual screening studies on Mpro have already been published, and in these studies, nanomolar-level non-covalent inhibitors have been identified (J. Am. Chem. Soc. 2022, 144, 2905−2920; J. Med. Chem. 2022, 65, 6499−6512; ACS Med. Chem. Lett. 2023, 14, 10, 1434–1440). I am not sure about the significance of the current authors searching for analogs of inhibitors for other coronaviruses for virtual screening. Moreover, there has been no in-depth SAR studies conducted on the identified hits to obtain nanomolar inhibitors, and there are no experimental structures to validate the docking poses.
Our group has successfully identified enzymatic inhibitors from fingerprint-based similarity searching (references 71, 72). Thus, we decided to apply the same strategy to the identification of Mpro inhibitors. The fact that the identified compounds have low/middle potency could not be known before completing our study.
The lack of experimental structures is one of the limitations of this study, especially after knowing that the docking pose of compound 13c is not stable in molecular dynamics simulations presented in Figure S3 (experiments requested by a different Reviewer). We have pointed out that experimental structures would be required for further development of similar compounds (lines 265-266).
Comment 2: Could the authors summarize the reported antiviral activity of reference compounds with the IC50/EC50 values?
We have added Table S2 to report the IC50/EC50 of compounds within the reference group with viral protease inhibitory activity.
Comment 3: Is there any correlation between the docking scores and the predicted antiviral activity? In this manuscript, I did not find information regarding the docking scores.
We have included Table S4 to report the docking scores of compounds 13-13c. We have made explicit the lack of correlation between docking scores and experimental activities in the revised manuscript (lines 254-255). However, a correlation between docking score and activity in a complex system is not always expected. Multiple docking protocols for Mpro have shown below-average performance (ref 39, 78), indicating that the inherent limitations of molecular docking and the complexity of the biological target may not accurately capture the molecular interactions that influence the observed activity. We are discussing this in detail in lines 259-263.
Comment 4: Regarding the docking poses of compounds 13-13c, in Figure 4, it is evident that in all compounds, the poses show pyridone within the S2 pocket forming π-π interactions with His41. However, in Figure S2, this seems to be inconsistent, especially in Figure S2(A). Could the authors please provide an explanation for this discrepancy?
Thank you for bringing this to our attention. After double-checking, we found that the 2D representations were not made with the best-ranked pose. Consequently, we have corrected figure S2 in the revised version.
Reviewer 3 Report
Comments and Suggestions for Authors
Manuscript Number: pharmaceuticals-2846685
Type of manuscript: Article
Title: Identification of SARS-CoV-2 main protease inhibitors by chemical similarity analysis combined with machine learning
The article investigates the potential of SARS-CoV-2 Main Protease (Mpro) as a target for anti-SARS-CoV-2 drug development. Through large-scale virtual screening and machine learning algorithms, the study identifies eighteen computational hits with high structural diversity and drug-like properties. Compound 13 (ZINC13878776) emerges as a significant inhibitor of SARS-CoV-2 Mpro activity, leading to the evaluation of its structural analogues (13a, 13b, and 13c) as Mpro inhibitors. These compounds exhibit in vitro antiviral activity, suggesting their potential as lead compounds for the development of new Mpro inhibitors against anti-SARS-CoV-2.
While I appreciate the thoroughness of the study, I would like to suggest a clarification regarding the strategy employed in selecting a group of 69 compounds from the literature, known for their activity in inhibiting coronavirus growth. It is noted that these compounds may have different mechanisms of action. The authors use this set of compounds as a basis for identifying structurally similar ones and assume that these will act as Mpro inhibitors. It strikes me as an unconventional approach to choose compounds with potentially diverse mechanisms of action, especially considering the focus on Mpro inhibition. A more conventional starting point might involve compounds already recognized as Mpro inhibitors. Although the strategy ultimately proved successful in identifying a compound that was experimentally confirmed to inhibit Mpro, it would be beneficial to clarify this particular aspect in the article.
The introduction could be refined by incorporating information on the specific residues of MPro that interact with known inhibitors. This information becomes crucial later in the manuscript when discussing the predicted interactions of the tested compounds.
The authors use IC50 values to represent the inhibitory concentration of the compounds against Mpro and when assessing the inhibitory effect on viral growth. While IC50 is widely accepted for quantifying enzyme inhibition, it might be beneficial to use the term EC50 when assessing the inhibitory effect on viral growth. In the context of antiviral activity, particularly in plaque reduction assays, EC50 (effective concentration) is a more suitable term. EC50 is applied in dose-response curves exhibiting an upward trend, for example when representing the cell growth in presence of the virus and at increasing inhibitor concentrations. The term "effective concentration" aligns better with the outcomes of assays that measure the impact of inhibitors on viral replication. IC50 (inhibitory concentration) is used for dose-response curves that go downhill, such as when representing the Mpro activity at increasing inhibitor concentrations, and should be used to represent the inhibitory concentration of the compounds against Mpro.
Minor corrections:
- Lines 51-53: If you want to specify that both the papain-like protease (PLPro) and the Mpro (also known as 3CLPro) are involved in the release process, you would use "and" in the sentence. Therefore, the revised sentence would be: "Generation of NSPs from pp1a and pp1ab requires the release by the papain-like protease (PLPro) and the Mpro (also known as 3CLPro) by self-cleavage [1,6,7]."
- Lines 56-57: Mpro and PLPro does not participate in the proofreading activity of the RdRp. They only participate in the excision of the polyproteins 1a and 1ab.
- Table 1 presents values that are predictions. I recommend that the authors explicitly state in the manuscript that the values in Table 1 are predicted values.
Author Response
We thank the Reviewer for her/his comments and suggestions. Please find below the detailed response to the observations made.
1. While I appreciate the thoroughness of the study, I would like to suggest a clarification regarding the strategy employed in selecting a group of 69 compounds from the literature, known for their activity in inhibiting coronavirus growth. It is noted that these compounds may have different mechanisms of action. The authors use this set of compounds as a basis for identifying structurally similar ones and assume that these will act as Mpro inhibitors. It strikes me as an unconventional approach to choose compounds with potentially diverse mechanisms of action, especially considering the focus on Mpro inhibition. A more conventional starting point might involve compounds already recognized as Mpro inhibitors. Although the strategy ultimately proved successful in identifying a compound that was experimentally confirmed to inhibit Mpro, it would be beneficial to clarify this particular aspect in the article.
Our strategy followed a two-step process. As the Reviewer noticed, in our first step we employed references with multiple activities. This has been specified in the revised version of the manuscript (line 133). As result of this primary screening, we obtained a list of commercially available compounds that may have anti-coronavirus activity independently of its mechanism of action. This has been explained in “results and discussion” (lines 136-137). In the second step, we searched for compounds with potential inhibitory activity against Mpro. This allowed us to focus on our study. We have clarified this information in lines 146-149.
2. The introduction could be refined by incorporating information on the specific residues of MPro that interact with known inhibitors. This information becomes crucial later in the manuscript when discussing the predicted interactions of the tested compounds.
We have incorporated the requested information in the introduction (lines 84-87).
3. The authors use IC50 values to represent the inhibitory concentration of the compounds against Mpro and when assessing the inhibitory effect on viral growth. While IC50 is widely accepted for quantifying enzyme inhibition, it might be beneficial to use the term EC50 when assessing the inhibitory effect on viral growth. In the context of antiviral activity, particularly in plaque reduction assays, EC50 (effective concentration) is a more suitable term. EC50 is applied in dose-response curves exhibiting an upward trend, for example when representing the cell growth in presence of the virus and at increasing inhibitor concentrations. The term "effective concentration" aligns better with the outcomes of assays that measure the impact of inhibitors on viral replication. IC50 (inhibitory concentration) is used for dose-response curves that go downhill, such as when representing the Mpro activity at increasing inhibitor concentrations, and should be used to represent the inhibitory concentration of the compounds against Mpro.
We have changed figures and text to include the suggested corrections. Now, Mpro activity graphs go downhill (figure 3b) and the manuscript uses “EC50” when referring to the activity of the compounds in plaque reduction assays (figure 5).
4. Minor corrections.
- Lines 51-53: If you want to specify that both the papain-like protease (PLPro) and the Mpro (also known as 3CLPro) are involved in the release process, you would use "and" in the sentence. Therefore, the revised sentence would be: "Generation of NSPs from pp1a and pp1ab requires the release by the papain-like protease (PLPro) and the Mpro (also known as 3CLPro) by self-cleavage [1,6,7]."
- Lines 56-57: Mpro and PLPro does not participate in the proofreading activity of the RdRp. They only participate in the excision of the polyproteins 1a and 1ab.
- Table 1 presents values that are predictions. I recommend that the authors explicitly state in the manuscript that the values in Table 1 are predicted values.
The suggested corrections have been incorporated to the revised version in lines 57, 61-62, and Table 1.
Reviewer 4 Report
Comments and Suggestions for Authors
Manuscript pharmaceuticals-2846685 entitled “Identification of SARS-CoV-2 main protease inhibitors by chemical similarity analysis combined with machine learning” is a wonderful study of much interest to the med. chem. community. The authors should just incorporate minor changes below to elevate even further the quality of their work.
Line 77; “However, there is still a need to find new inhibitors for SARS-CoV-2 Mpro” Indeed there are academic efforts you can cite for the benefit of the reader, for example: https://doi.org/10.3390/molecules25245808 and https://doi.org/10.3390/molecules26103003.
Line 88, “Thus, VS reduces calculation time and storage space” Not necessarily as (HT)VS can be done with expensive docking protocols. But it is flexible ever enough and some simple filtration approaches can be performed. (You chose a quick VS screening approach but this is not general to all HTVS approaches.)
Line 122; You write You have 17 M initial library composed from ChEMBL, ENAMINE REAL, etc. etc. – If You combined those libraries, you would have >> 17 M. How did You arrive at 17 M initial library. Please describe in detail (materials and methods) even if You cite reference [57]. This could be useful for the reader.
Line 130-133: Can You describe for the reader what these machine learning algorithms actually do or look for? The reader is a bit confused with the step of reduction from 4k to 18 compounds! Also: “finding compounds against SARS-CoV-2”. Are the algorithms employed in consensus or one or another…. Are they tailored for 3CLpro or SARS-CoV-2 activity in general?
Figure 1: Why similitude in particular?
Lin 146: “The predicted physicochemical, pharmacokinetic, and toxicological properties for all computational hits were in the range of typical drug-like molecules, which was expected, given the composition of the screening library.” How were they predicted? How is the composition of the library related? Describe for the reader.
So you sought after Lipinski conformity and oral bioavailability. Is this sensible in this stage of design? Please elaborate on this a bit more.
Figure 2: Please reference GC376. https://doi.org/10.1038/s41467-020-18233-x
Line 166: What about compound 7? What were its problems?
Line 178: “Our analysis indicates that compound 13c may have an extended half-life in humans (Table 1).” How, which software?
Materials and methods. Computational protocols should be described in just a bit more detail so the reader can repeat or get an idea of how to conduct a similar approach.
Chapter 2.3. SAR expansion is difficult with 4 compounds. Please rephrase; analog search or something similar. You did not postulate any SAR at this stage so do not use SAR expansion for this stage. Alos SAR study in general cannot be conducted with 4 compounds.
Table 1: Mention predicted and Software employed.
2.4. Mention software, despite the methods section.
Author Response
We thank the Reviewer for her/his comments and suggestions. Please find below the detailed response to the observations made.
1. Line 77; “However, there is still a need to find new inhibitors for SARS-CoV-2 Mpro” Indeed there are academic efforts you can cite for the benefit of the reader, for example: https://doi.org/10.3390/molecules25245808 and https://doi.org/10.3390/molecules26103003.
We have added the suggested references (49 and 50).
2. Line 88, “Thus, VS reduces calculation time and storage space” Not necessarily as (HT)VS can be done with expensive docking protocols. But it is flexible ever enough and some simple filtration approaches can be performed. (You chose a quick VS screening approach but this is not general to all HTVS approaches.)
The text has been modified to remove the idea that all VS are fast to perform. The text still states that VS requires shorter time than experimental high-throughput screening (lines 96-99).
3. Line 122; You write You have 17 M initial library composed from ChEMBL, ENAMINE REAL, etc. etc. – If You combined those libraries, you would have >> 17 M. How did You arrive at 17 M initial library. Please describe in detail (materials and methods) even if You cite reference [57]. This could be useful for the reader.
We have added further details to section 3.1 to clarify the procedure for library construction.
4. Line 130-133: Can You describe for the reader what these machine learning algorithms actually do or look for? The reader is a bit confused with the step of reduction from 4k to 18 compounds! Also: “finding compounds against SARS-CoV-2”. Are the algorithms employed in consensus or one or another…. Are they tailored for 3CLpro or SARS-CoV-2 activity in general?
We have included further information on the employed ML algorithms and declared that we selected compounds complying with both strategies (section 3.2).
5. Figure 1: Why similitude in particular?
To clarify this point we included in the figure 1 caption that the similarity analysis was performed with the Tanimoto coefficient and three structural fingerprints.
6. Lin 146: “The predicted physicochemical, pharmacokinetic, and toxicological properties for all computational hits were in the range of typical drug-like molecules, which was expected, given the composition of the screening library.” How were they predicted? How is the composition of the library related? Describe for the reader.
We made a mistake in that sentence; thank you for pointing it out. Sentence has been modified in the manuscript (lines 165-166).
7. So you sought after Lipinski conformity and oral bioavailability. Is this sensible in this stage of design? Please elaborate on this a bit more.
We have elaborated on the importance of the predicted pharmacokinetic properties at an early stage of drug discovery (line 109-111).
8. Figure 2: Please reference GC376. https://doi.org/10.1038/s41467-020-18233-x
The original report discovering the activity of GC376 as Mpro inhibitor is quoted (reference 18).
9. Line 166: What about compound 7? What were its problems?
We did not perform further experiments with compound 7 since it showed high variability among replicates. As result, the Mpro inhibition by compound 7 was not statistically significant. No changes were made to the manuscript.
10. Line 178: “Our analysis indicates that compound 13c may have an extended half-life in humans (Table 1).” How, which software?
The information has been added to Table 1 heading.
11. Materials and methods. Computational protocols should be described in just a bit more detail so the reader can repeat or get an idea of how to conduct a similar approach.
Our new version has improved the description of the employed methods.
12. Chapter 2.3. SAR expansion is difficult with 4 compounds. Please rephrase; analog search or something similar. You did not postulate any SAR at this stage so do not use SAR expansion for this stage. Alos SAR study in general cannot be conducted with 4 compounds.
We have rephrased “SAR expansion” along the manuscript.
13. Table 1: Mention predicted and Software employed.
Table 1 has been modified as suggested.
14. 2.4. Mention software, despite the methods section.
We have included in the description of the results the employed software.
Round 2
Reviewer 2 Report
Comments and Suggestions for Authors
The author has resolved and answered my question. I have no further comments.